# MAGICAL: A multi-class classifier to predict synthetic lethal and viable interactions using protein-protein interaction network

**Anubha Dey[1], Suresh Mudunuri[2], Manjari Kiran[1]***

**1** Department of Systems and Computational Biology, School of Life Sciences, University of Hyderabad, Hyderabad, India, **2** Centre for Bioinformatics Research, SRKR Engineering College, Andhra Pradesh, India

* manjari.hcu@uohyd.ac.in

## Abstract

Synthetic lethality (SL) and synthetic viability (SV) are commonly studied genetic interactions in the targeted therapy approach in cancer. In SL, inhibiting either of the genes does not affect the cancer cell survival, but inhibiting both leads to a lethal phenotype. In SV, inhibiting the vulnerable gene makes the cancer cell sick; inhibiting the partner gene rescues and promotes cell viability. Many low and high-throughput experimental approaches have been employed to identify SLs and SVs, but they are time-consuming and expensive. The computational tools for SL prediction involve statistical and machine-learning approaches. Almost all machine learning tools are binary classifiers and involve only identifying SL pairs. Most importantly, there are limited properties known that best describe and discriminate SL from SV. We developed MAGICAL (*Multi-class Approach for Genetic Interaction in Cancer via Algorithm Learning*), a multi-class random forest based machine learning model for genetic interaction prediction. Network properties of protein derived from physical protein-protein interactions are used as features to classify SL and SV. The model results in an accuracy of ~80% for the training dataset (CGIdb, BioGRID, and SynLethDB) and performs well on DepMap and other experimentally derived reported datasets. Amongst all the network properties, the shortest path, average neighbor2, average betweenness, average triangle, and adhesion have significant discriminatory power. MAGICAL is the first multi-class model to identify discriminatory features of synthetic lethal and viable interactions. MAGICAL can predict SL and SV interactions with better accuracy and precision than any existing binary classifier.

## Author summary

Targeted therapy aims to selectively target cancer cells without damaging the normal ones. Synthetic lethality is a negative genetic interaction in which alteration of both genes leads to cell death and mediates drug sensitivity. In contrast, synthetic viability is a positive genetic interaction in which gene alteration rescues the cell sickness induced by alteration in the vulnerable gene and promotes cell viability, leading to drug resistance. Hence,

**Data Availability Statement:** The data and the codes are available at https://github.com/Anubhagithub/MAGICAL The database is available at http://sls.uohyd.ac.in/new/magicaldb

**Funding:** This work has been funded and supported by an IoE grant (IoE-RC2-21-012) from the University of Hyderabad. AD and MK acknowledge funding support from University of Hyderabad Institute of Eminence Grant (UoH-IoE-RC2-21-012). AD is a registered PhD student at the University of Hyderabad. MK also gratefully acknowledge the DBT BUILDER project and core grant support from the University of Hyderabad. The funders had no role in study design, data collection and analysis, decision to publish, or preparation of the manuscript.

**Competing interests:** The authors have declared that no competing interests exist.

identifying these genetic interactions is crucial to fostering selective treatment and improving the patient's health. We have designed MAGICAL, a multi-class classifier for predicting genetic interactions, a machine-learning model that can predict SL and SV based on the network properties. We aim to address how these genetic interactions get affected when the placement of the nodes (genes) in the network changes. As genetic interaction in cancer has a key role in precision oncology/targeted therapy, this work would enable researchers to understand how these interactions foster better treatment.

## Introduction

Targeted therapy aims to target proteins responsible for cancer cells' growth, improving patients' physical and mental health [1]. With the revolution in transcriptomics, proteomics, and metabolomics data, patients can be suggested with treatment that can enhance their health and improve survival. Some approaches that offer selective treatment are based on genetic interaction between gene pairs. Genetic interaction is the phenotypic outcome resulting from two or more gene interactions. Synthetic Lethality (SL) and Synthetic Dosage Lethality (SDL) are types of negative genetic interaction in which inhibition/mutation in either of the genes does not affect the cancer cell survival, but the inhibition in the partner gene makes the cell lethal/sick [2,3]. Synthetic Viability (SV) is a positive genetic interaction in which the inhibition of one gene makes the cancer cell sick, while the inhibition of the partner gene rescues the effect and promotes cell viability [4].

### Identification of genetic interactions

The experimental approaches to identify genetic interactions include low-throughput independent studies involving a few genes, such as TP53, RAS, and KRAS [5,6]. Sh-RNA and RNAi-based techniques have so far identified very few SL interactions. Such techniques are low throughput and are often associated with off-target effects. Yeast, as a eukaryotic model organism, has been used to identify genetic interactions in humans. Yeast and humans are two distinct species, so mapping the orthologs is unreliable, and thus, identifying orthologous genetic interactions may not be correct [7]. Recent techniques include CRISPR for successfully retrieving SL and SV pairs, but the major limitation of CRISPR-based technology is that it is expensive and time-consuming [8]. Most experimental techniques are laborious, cost-ineffective, and time-consuming, and thus, there is a need for computational prediction.

### Computational techniques to identify genetic interactions

Several statistical methods have been reported to identify genetic interactions in cancer utilizing mutation, copy number alteration, and gene expression [9–13]. Statistical tools are mostly based on assumptions on the biological dataset. They are not trained on the experimentally known pairs. These models fail to find the non-linear correlations between the dependent and independent variables. Statistical models are also often unable to address the problem of dimensionality reduction and require the incorporation of machine learning. Different machine-learning models are available that enable the successful prediction of SL pairs [14–20]. The existing machine learning tools are mostly limited to SL prediction and are binary class models. These binary classifiers predict only SL interactions and classify them with NOT (SL/ NOT SL). Although SV interactions have been under-explored, a recent study by Liu *et al.*, predicts SV interactions by building a binary ensemble classifier [21].

### The Gap in the field and the need for a multi-classification model

As mentioned earlier, all the machine learning models in the literature are restricted to binary classification, mostly for predicting SL pairs. We believe an ideal genetic interaction prediction can classify positive and negative interactions [22].

As existing models categorize the data into SL and NOT SL, the NOT SL dataset might contain genetic interactions that are SVs, SDLs, etc. So, there is a need for a robust, computationally efficient, and interpretable model whose determinant features help decipher the differences between the different classes of genetic interactions. To our knowledge, no model performs a multi-class prediction for identifying different classes of such interactions. The study by Wang *et al.* demonstrates that most of the previous models are limited to imbalanced data, fail to identify informative features, are less decipherable, and are more like a black box [23].

### Network properties as features in the multi-class model

Network properties have been extensively used for predicting genetic interaction between Yeast and humans [24,25]. A study by Talavera *et al.* in 2013 identified that protein products of SL pairs interacting physically are highly conserved [26]. This unravels the idea of identifying many SL pairs utilizing the physical protein-protein interaction network. Network properties have never been used to predict SV pairs. Drawing insights from the mentioned literature, we employ network properties as a feature in our multi-class model. In the present study, we have developed **MAGICAL (Multi-class Approach for Genetic Interaction in Cancer via Algorithm Learning),** a multi-class classifier that allows one to understand the differences between negative and positive interaction. With MAGICAL, we aim to address the following questions? How are SL and SV pairs placed in the physical protein-protein interaction network? Do the protein products of SV pairs also physically interact? How different are these pairs in terms of their network properties?

MAGICAL is trained on 20 network properties, among which properties like shortest path, average neighbor2, average betweenness, average triangle, and adhesion serve as determinants in classifying SL and SV interactions. We notice that SL pairs with a higher value of shortest path are farther in the network than SVs. SL pairs have a higher degree and betweenness values, indicating that they are more central in the network. SL interactions form more communities, engage in more crosstalk, and are in different modules, whereas SV pairs are closer to each other, placed in similar modules, and involved in less crosstalk. MAGICAL outperformed existing binary classifiers and also predicted novel SL and SV interactions.

## Method and materials

The steps involved in the development of MAGICAL are shown stepwise in Fig 1.

### Training and validation dataset

The SL and SV pairs are retrieved from CGIdb, a Cancer Genetic Interaction database; Bio-GRID, the Biological General Repository for Interaction Datasets; and SynLethDB (synthetic lethality database), a repository for SL and SV interactions [27–29]. A total of 26,445 SL, 3,867 SV, and 9,986 NOT interactions are obtained after data preprocessing and mapping the pairs to the network. The performance of MAGICAL has also been compared with a binary classifier SLant that is trained on features derived from physical protein-protein interaction network [25]. SLant has been trained on the BioGRID dataset where a pair is labeled as SL if the respective gene of the protein pair has a negative interaction as reported in BioGRID and non-SL if

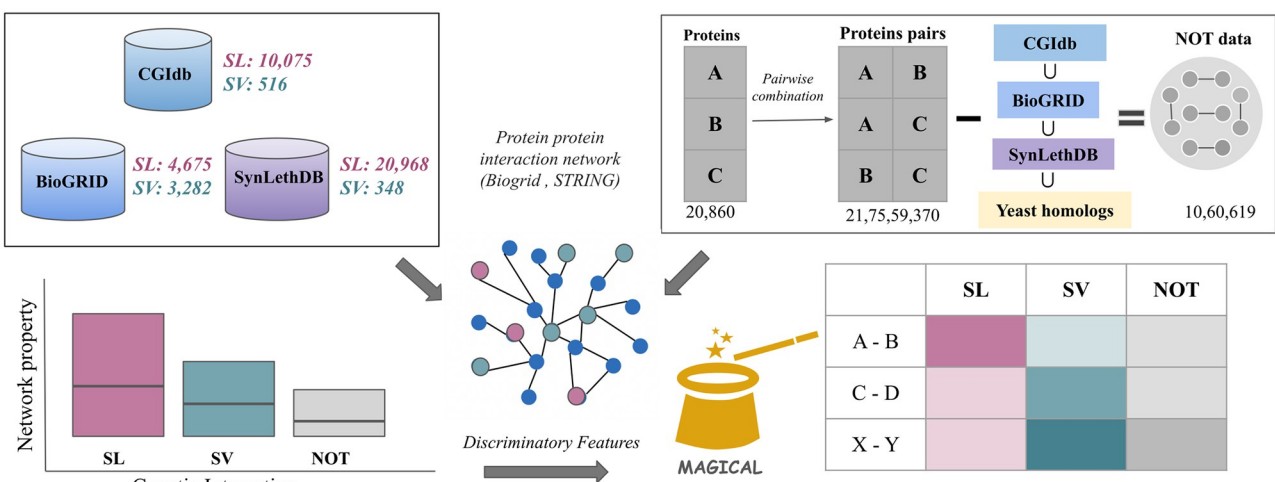

**Fig 1. The SL and SV pairs are retrieved from three sources: CGIdb, BioGRID, and SynLethDB.** Further, these pairs are mapped to the physical protein-protein interaction network, and the network properties are calculated. After identifying the determinant features, MAGICAL is built to classify a given gene pair as SL, SV, or NOT. The NOT dataset is built utilizing three steps: first, retrieval of all proteins from the BioGRID dataset and generating pairwise combinations. Second, removal of the pairs present in CGIdb, BioGRID, and SynLethDB, along with yeast orthologs. Third, Random picking of ~10,000 pairs is utilized for the model building.

the pair is not reported in the BioGRID data. Due to the unavailability of the SLant model, we developed a binary class model (SLant) utilizing network property values (calculated by us for BioGRID protein-protein interaction network) for the SL pairs reported in BioGRID. SLant, as mentioned in Benstead *et al.*, has been trained on BioGRID data consisting of 411 SL and 411 non-SL interactions [25]. We retrieved 411 SL pairs from BioGRID and 411 non-SL from our 'NOT' data. The determinant network properties as features (coreness, adhesion, and cohesion) mentioned by Benstead *et al.*, are utilized in training this binary class model (SLant) [25]. We compare the accuracy of this binary model (SLant) to that of the MAGICAL, which is trained on five determinant network properties (shortest path, average neighbor2, average betweenness, average triangle, and adhesion) and establish the efficiency of MAGICAL over SLant.

In addition to this, different other datasets are also considered to validate the model. These include TCGA, DepMap, and CRISPR-related individual study datasets [30–32]. The DepMap dataset consists of genome-wide CRISPR loss-of-function screens in cell lines that are genomically characterized. DepMap also consists of drug-gene and gene-gene combinatorial CRISPR screens. DepMap serves as a reference map that connects tumor features with tumor dependencies. Similar to DepMap, one of the published data utilizing CRISPR has also been used. The authors performed a genome-wide CRISPR screening of ~17,000 protein-coding genes and checked with the inhibition of ATR whether or not the cancer cells are lethal or viable [32]. A negative z-score value depicts drug sensitivity (SLs), while a positive z-score represents drug resistance (SVs). Kaplan Meier curves have been plotted using cBioportal with TCGA data consisting of 2683 samples from 2565 patients [33].

## The NOT dataset

MAGICAL is trained for three classes: SL, SV, and NOT. The NOT dataset is constructed with a multi-step approach. A pairwise combination of all the proteins listed in BioGRID is generated. The reported experimental genetic interactions from BioGRID, CGIdb, and SynLethDB

are removed along with the removal of human pairs, the homologs of which are known to be genetically interacting in yeast. Further, 10,000 pairs are randomly selected as NOT dataset for model building. This step is repeated 1000 times to develop 1000 models to avoid bias generated due to one NOT dataset. The accuracy of models, however, doesn't change much upon changing the NOT dataset (S1 Fig).

## Balancing the imbalance dataset

To balance the imbalance in the dataset, the "Synthetic Minority Oversampling Technique" SMOTE package is employed. SMOTE is an R library that handles the imbalance in the input data by either oversampling, undersampling, or both. Random under-sampling has been utilized here, where the majority class is under-sampled, and the different classes are balanced accordingly. SMOTE balances the imbalanced data, resulting in 11,238 SL, 11,601 SV, and 4,230 NOT data, totalling 27,069 pairs. This data is divided into a 70–30 ratio, in which the training dataset includes 70% (18,948) of the genetic interactions, and the remaining 30% (8,121) serves as the test data.

## Network analysis

The physical protein-protein interaction data is downloaded from BioGRID, and a network of physically interacting proteins is constructed for the MAGICAL-core model. Experimental and predicted physical protein-protein interactions from the STRING database have been used for the MAGICAL-combined model [34]. The self-loops and duplicate interactions are discarded, and only the unique interactions are considered. The network is analyzed using "igraph", an R package, to calculate the network properties. A total of 20 node and pairwise properties are calculated and listed in Table 1.

**Table 1. List of network properties used as features.**

| Network Property | Description |
|---|---|
| Degree | The total number of connections of a node |
| Betweenness | The extent to which the node of interest mediates the interconnection of other nodes |
| Closeness | A measure of how close a node is to the remaining nodes |
| Coreness/K-Core | The number of connected components that remain after removing all vertices with degree k |
| Constraint | Measures the extent of how much a node's connection is invested in a single cluster of neighbors |
| Eccentricity | Calculated as the reciprocal of the maximum of shortest path lengths from that node to all other nodes |
| Eigen-centrality | Measures the importance of a node that is connected to at least one hub in the network |
| Hub-score | The number of well-connected hubs to which the node of interest is linked. |
| Neighborhood n size | Set of all vertices not farther than that vertex in the network |
| Triangle | A set of three nodes where each node has a relationship with the other two, referred as 3-cliques |
| Common-neighbor | A vertex/node is a neighbor of another vertex, or two vertices are adjacent if are incident to the same edge. |
| Community-detection | Yields set of connected communities, where subsets of all communities are assigned optimally |
| Shortest-path (pairwise) | The minimum path length traversed from the source to the destination node |
| Cohesion (pair-wise) | The minimum nodes that might be removed to result in two separate sub-graphs that separate the source and the destination nodes |
| Adhesion (pair-wise) | The minimum number of edges that might have severed to result in two separate sub-graphs that separate the source and the destination nodes |

The downloaded SL and SV pairs from different sources are mapped to the physical protein-protein interaction network. The average of topological properties of the two genes in a pair is calculated. For example, if a and b are a SL pair. The network property (degree) for the pair is calculated as

$$\text{degree(ab)} = (\text{degree(a)} + \text{degree(b)})/2$$

The two properties' average instead of summation, difference, maximum, and minimum have been considered. This is based on the fact that there is no difference in accuracy, precision, recall, and F1 score received on average compared to the rest (S2 Fig).

Apart from the basic topological properties, such as degree, betweenness, closeness, coreness, etc., we employed attributes like community detection, common neighbors, triangles, etc. For detecting the communities, we used the *cluster_leiden* method, "CPM" and "Modularity" objective functions, and also employed a resolution parameter of 0.005. Leiden yields connected communities with better partition and is time-inexpensive than other community detection algorithms like Louvain and Walktrap [35]. The *distances* function is utilized for calculating the shortest path, and the infinite value for the shortest path is replaced with the network's diameter. For the calculation of cohesion and adhesion, *vertex.connectivity* and *edge.connectivity* functions have been used respectively.

## Selection of ML model

Different supervised learning approaches are incorporated: Random Forest, Decision Trees, K-Nearest-Neighbor, Naive Bayes, and Deep Learning. The random forest outperformed all the classifiers with an average accuracy of ~81.00% utilizing 10 cross-validations (S3 Fig).

## Feature extraction

The *varimp* function of the randomForest package identifies the determinant features. A total of 1000 bootstrapping is carried out to select the discriminatory features. The NOT data is randomly picked for every model generated, and the entire dataset is undersampled each time. For every model run, training data changes each time. The *varimp* functionality is such that each feature is picked once, and instead of the actual values, the number of entries for this variable is permuted and fed to the model. After every model run, the accuracy is noted; if the accuracy drops to a greater extent, the attribute/feature is considered to be of utmost importance. The permuted variable for which the accuracy drops the most is regarded as the most important (S4 Fig). Features are ranked based on the frequency of their occurrence in all 1000 models. We have also performed 100 bootstrapping to check for the variable importance. The top five determinant properties remain the same for 100 bootstraps as well (S5 Fig). In order to ensure that no two features are highly correlated, a correlation plot is generated to identify the correlation of these properties. Since most network properties are correlated, it increases the chance of overfitting and does not allow the model to learn variety, further affecting model accuracy. Properties like shortest path and adhesion are not correlated with the rest of the network properties considered for model training (S6 Fig). Although average neighbor2, average betweenness, and average triangle are positively correlated, dropping any one reduces the model accuracy to a greater extent and is retained in the feature list (S7 Fig).

## Gene ontology

The basic version of the Gene Ontology (GO) and the gene ontology annotation file are retrieved from the gene ontology database [36,37]. After pre-processing of the data, GO terms

for each gene are obtained. For the entire analysis, GO terms for biological processes have been considered. Average and Jaccard Index of GO terms have been calculated for SL, SV, and NOT pairs.

The average of GO terms for a pair is the summation of the number of GO terms for both genes divided by 2

$$\text{Average(ab)} = \text{number of GO term of } (a + b)/2$$

The Jaccard Index (JI) is the division of intersection to the union of GO terms for both genes

$$\text{JI(ab)} = \text{Intersection of GO terms of a and b /Union of GO terms of a and b}$$

### Statistical analysis

Different tests, such as the Kolmogorov-Smirnov (KS) test and the DeLong test, have been performed to provide statistical significance to the analysis carried out in the study. The KS test has been carried out to test the differences in the distribution of network property values for all three classes: SL, SV, and NOT. The DeLong test is performed to test the difference in the accuracy between the two models. We employed the *ks.test* function in R to establish the significant differences in the distribution of the genetic interactions. For the DeLong test *roc.test* function in R with the "delong" method is used.

## Results

### Genetic interactions can be classified based on network properties

Network properties of physical protein-protein interactions network are used to classify the genetic interactions. Among the network properties in Table 1, the discriminatory features are picked based on bootstrapping on 1000 models (*please refer to Materials and Methods*) (Fig 2). We selected the features based on two criteria: i) Features selected in >90% models ii) Features ranked at least the top 5 based on drop in accuracy upon permutation of the feature values. The network properties that most frequently affect the model's accuracy in classifying SL, SV, and NOT are the shortest path, average neighbor2, average betweenness, average triangle, and adhesion (Fig 2). Among the pair-wise properties, shortest path, which is the minimum number of edges between the pairs, and adhesion, the minimum number of edges removed to separate pairs into two sub-networks, play an important role in classifying SL, SV, and NOT. Average neighbor2, average betweenness, and average triangle are a few node-wise properties linked to high centrality measures of nodes that are also selected by > 90% models and affect accuracy if removed from the feature list. MAGICAL comprises these five above-mentioned features to classify SL, SV, and NOT. Interestingly, average eigen-centrality and average hub-score are not picked by any model and are unsuccessful in classifying genetic interactions.

Intriguingly, the partners of SV pairs are closer compared to the SL pairs (p-value < 2.2e-16, KS test), which is also far from the NOT pairs (Fig 3). SL pairs have the least adhesion values, indicating that fewer edges are needed to place the two nodes into separate subgraphs than SV and NOT (p-value < 2.2e-16, KS test). SL pairs have a higher value of average neighbor2, average betweenness, and average triangle than those of the SVs and NOT, where the former has higher values than the latter (p-value 4.87e-06, < 2.2e-16, and 1.65e-06 respectively, KS test).

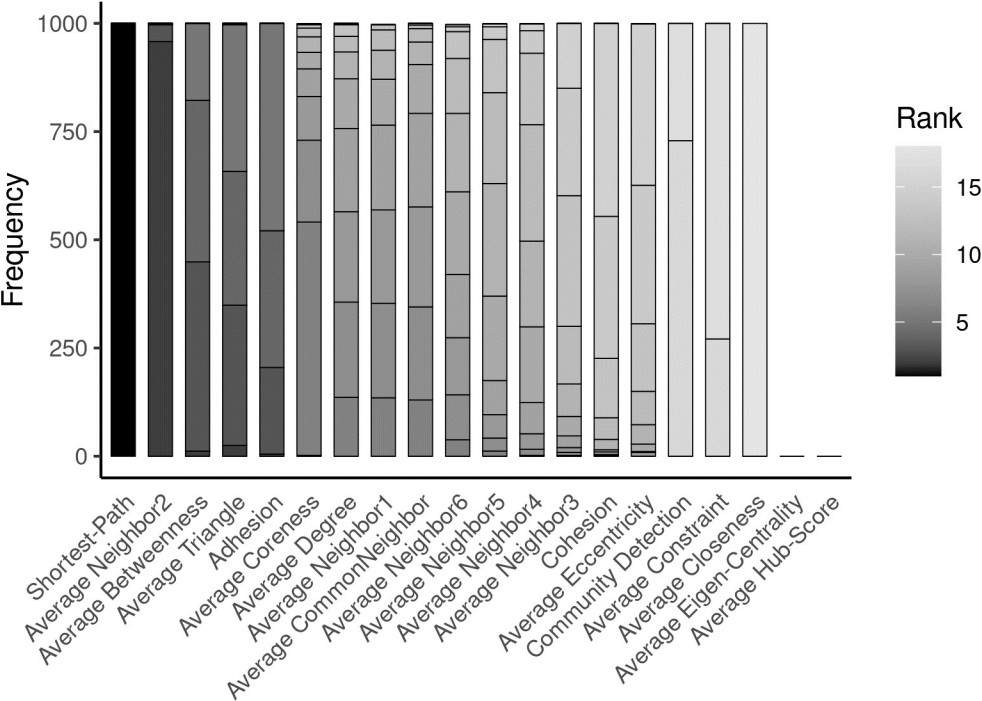

**Fig 2. A stacked barplot to represent the determinant features that contribute to building MAGICAL.** The discriminatory features are identified by 1000 bootstraps and counting the number of times each model chooses a property.

## MAGICAL shows ~80% accuracy with only five network properties

The training dataset includes 70% of the genetic interactions from CGIDB, BIOGRID, and SynLethDB, along with the NOT data, and the remaining 30% serves as the test data [27–29]. MAGICAL-core model (features values obtained from experimental protein-protein interaction network) reaches an accuracy of ~81% with 0.897–0.913 at 90% confidence interval. The accuracy ranges from 84.57% to 75.71% with a running point from 0.1 to 0.9 (S8 Fig), showing a high true positive rate and low false positive rate for different running points. The network properties "shortest path," "average neighbor2," "average betweenness," "average triangle, and "adhesion" serve as discriminatory features. The prediction accuracies is depicted as how many of the actual number of classes are correctly predicted as SL, SV, and NOT (Fig 4A). It is

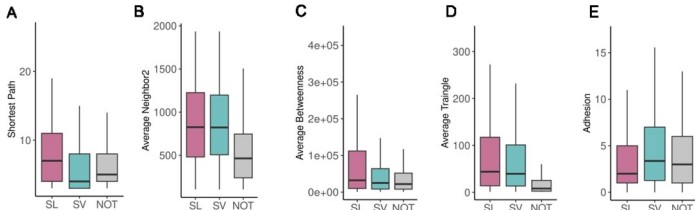

**Fig 3. Boxplots showing** A) SL interactions having higher value of shortest path than SVs (p-value < 2.2e-16, KS test). B) SL pairs having a higher value of average neighbor2 compared to SVs (p-value 4.87e-06, KS test). C) SL interactions having higher betweenness values to that of the SVs pairs (p-value < 2.2e-16, KS test). D) SL pairs having higher values of the average triangle, compared to SVs (p-value 1.65e-06, KS test). E) SL pairs having lower adhesion value than SV pairs (p-value < 2.2e-16, KS test).

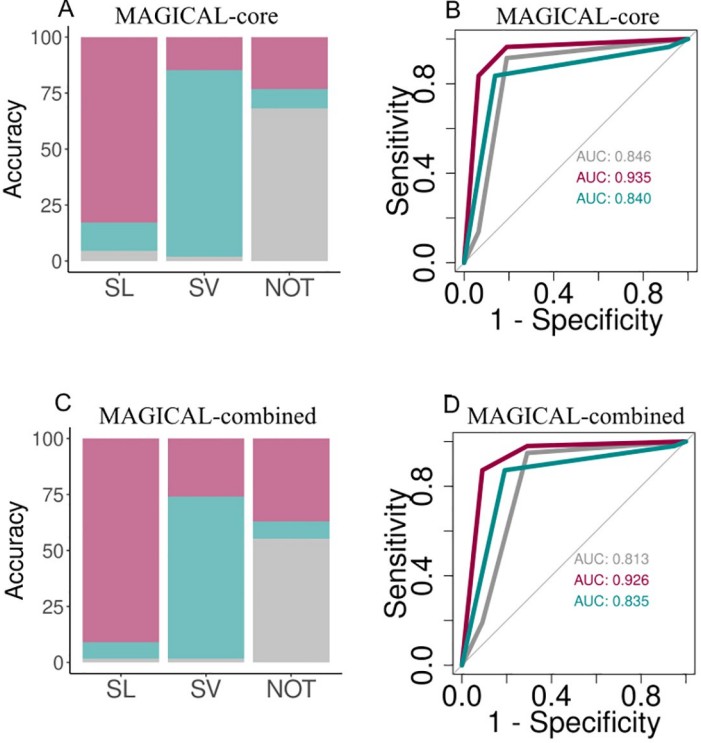

**Fig 4.** A) Stacked barplot representing the number of SL, SV, and NOT pairs correctly predicted by MAGICAL-core. B) The ROC plot depicting the performance of MAGICAL-core model. C) Stacked barplot representing the number of SL, SV, and NOT pairs correctly predicted by MAGICAL-combined. D) The ROC plot depicting the performance of the MAGICAL-combined model. The magenta, cyan, and grey colors represent SL, SV, and NOT.

also noted that the SL pair is predicted the best out of all the three classes (Fig 4B). The error rate versus the number of trees is plotted to test that MAGICAL is not overfitted. It is observed that the error rate first decreases and then saturates for more than 100 trees, indicating that increasing the number of trees does not impact model accuracy (S9 Fig).

For the MAGICAL-combined model (features values obtained from experimental + predicted protein-protein interaction network), the model results in an accuracy of 80% (Fig 4C). Similar to the MAGICAL-core model, the MAGICAL-combined model predicts SL pair better out of the three classes (Fig 4D).

## MAGICAL outperforms existing binary classifiers

As mentioned in the introduction, there is no multi-class model to predict genetic interactions; therefore, no comparison can be performed with multi-class classifiers. However, the previously reported binary classifier SLant is based on network properties and is ideal for comparing the performance. The genetic interactions reported in BioGRID is used to compare the performance of SLant and MAGICAL. MAGICAL not only outperforms in predicting SL pairs but also performs equally well for both balanced and unbalanced datasets (p-value 0.0003, < 2.2e-16, respectively, DeLong test). (Fig 5A and 5B).

The performance of MAGICAL is also compared with three binary classifiers, SL-NOT, SV-NOT, and SL-SV, trained on all 20 network properties. Interestingly, MAGICAL outperforms the binary classifiers with better AUC values (p-value 6.98e-06, 2.2e-16, and 2.2e-16, respectively, DeLong test). (S10 Fig).

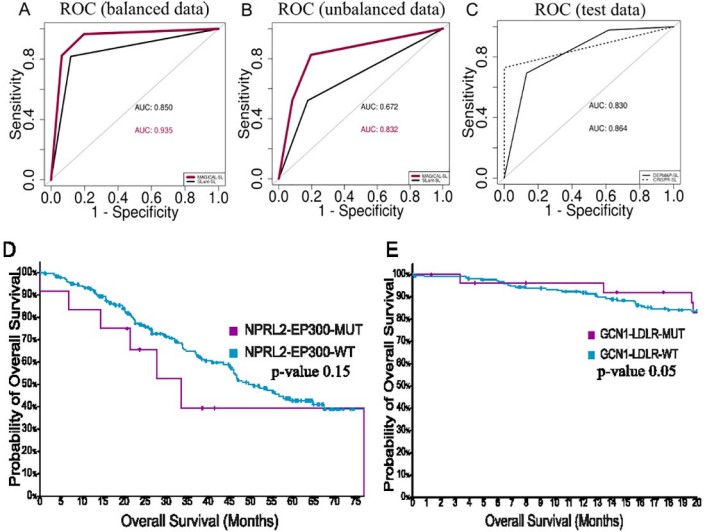

**Fig 5.** A) ROC curve showing an AUC of 0.850 and 0.935 for SLant-SL and MAGICAL prediction, respectively, for the balanced dataset (p-value 0.0003, DeLong test). B) For an unbalanced dataset, an AUC of 0.672 and 0.832 for SLant-SL and MAGICAL is respectively obtained (p-value < 2.2e-12, DeLong test). C) ROC curve showing the prediction accuracy of 0.83 and 0.86 for DepMap and Wang *et al.*, data, respectively. D, E) Kaplan Meier curve showing SL Pairs identified by SLant predicted as SVs by MAGICAL and vice-versa (p-value, 0.15 and 0.052 respectively, log-rank test).

## MAGICAL can predict SL pairs in independent datasets

To test MAGICAL performance on an independent dataset, known SL pairs are retrieved from DepMap (71,691) and predicted SV interactions from Gu *et al.*, (63) and Sahu *et al.* (813) [4,38]. After balancing the dataset, AUC value of 0.83 is obtained in predicting SL pairs reported in DepMap (Fig 5C). Similarly, a CRISPR-based experimentally identified SL and SV pairs are retrieved from a study performed by Wang *et al.* for the ATR gene in combination with ~17,000 protein-coding genes [32]. Surprisingly, an AUC value of 0.86 is observed in predicting SL, with an overlap of 4380 pairs out of 5656 SL pairs identified for ATR. Moreover, 4026 additional pairs are uniquely identified by MAGICAL listing probable SL pairs, which can be validated in future studies (S1 Table). Similarly, there is an overlap of 1173 SV pairs out of 5199 with CRISPR data and 1272 novel pairs are identified.

We also looked into a few other examples where SL pairs predicted by SLant are identified as SV pairs by MAGICAL and vice-versa. For example, NPRL2-EP300 has been identified as an SL gene pair by SLant; MAGICAL, on the contrary, predicts it as an SV pair. When checked against TCGA pan-cancer data, consisting of 2683 samples from 2565 patients, it is noted that the survival of patients with a mutation in both genes is worse than those with no mutations. Additionally, GCN1-LDLR, which is predicted as NOT a genetically interacting pair by SLant, MAGICAL classifies it as an SL pair. The survival of patients with mutations in both genes is better than that of patients with no mutation indicating SL pairs (Fig 5D and 5E).

MAGICAL could also identify some novel pairs. For example, IDH1 and PRKDC have been predicted as an SL pair with a prediction accuracy of ~70%. The literature demonstrates that the downregulation of IDH1 promotes tumor proliferation [39]. PRKDC, in contrast, is a DNA repair enzyme that repairs double-stranded breaks of damaged DNA. When IDH1 is downregulated or mutated, PRKDC, the SL partner of IDH1, repairs the damage and keeps the cell viable. But if PRKDC is inhibited with Ipilimumab, the damage cannot be repaired,

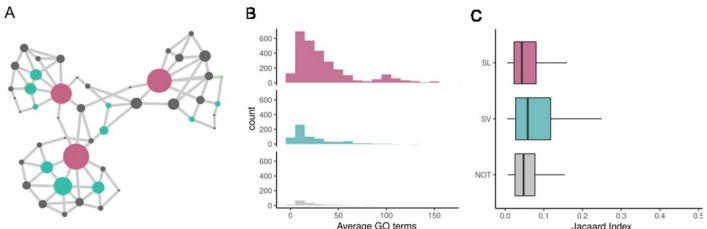

**Fig 6.** A) The position and placement of the SL (magenta) and SV (cyan) genes in the protein-protein interaction network. B) SL pairs have a higher value of average GO terms (p-value < 2.2e-16, KS test). C) SV pairs have a higher value of the Jaccard index (p-value 3.198e-12, KS test).

and the cell would undergo lethality reported in [40]. Similarly, FHL1 and ABL1 have been predicted as SV pair with a prediction accuracy of ~70%. ABL1 is an oncogene, so if ABL1 is mutated or downregulated, this would make the cancer cell sick, but if the partner FHL1 gene, which is a tumor suppressor, is inhibited, this inhibition will rescue the effect and promote cancer cell proliferation. This pair has not been studied and can be validated in further studies.

## SL pairs are distantly placed in different modules in the PPI network

SL pairs have a higher shortest path and lower adhesion, which suggests them being distantly placed from each other in the network. Interestingly, both SL and SV pairs are located in different communities (S11 Fig), but SL can be part of different subgraphs by adding a minimum number of edges. In contrast, SV pairs, being closer in the network, require more edges to be separated (Fig 6A). It is also observed that SL pairs have higher values of average GO terms compared to the SV pairs (p-value < 2.2e-16, KS test) (Fig 6B). However, SL pairs have a lower value of the Jaccard index than SV pairs, indicating that the SV pairs share more GO terms in common (p-value 3.198e-12, KS test) (Fig 6C).

For example, BRCA1 and VEGFA, an SL interaction, are associated with 60 and 167 GO terms, respectively. Surprisingly, a poor overlap of 3 GO terms indicates that this pair has a lot of unique GO terms of different and diverse roles. The common GO terms, "GO:0010628", "GO:0045766", and "GO:0045944" are associated with distinct biological roles, such as positive regulation of gene expression, positive regulation of angiogenesis, and positive regulation of transcription by RNA polymerase II, respectively. BRCA1 and VEGFA, although an SL pair, seem to be involved in two biological modules. Interestingly, for an SV interaction, MED12 and MED14, an overlap of 4 out of a total of 25 GO terms is observed. MED12 and MED14 are subunits of the mediator complex that regulate RNA polymerase II and function as a transcription coactivator. The GO terms "GO:0045944" and "GO:0060261" are associated with "positive regulation of transcription by RNA polymerase II". Thus, SV interactions tend to participate and share more biological processes than the SL pairs. In contrast, SL pairs reside in different modules, sharing fewer biological processes.

## MAGICAL database for easy data access

We have also deposited all the predicted SL and SV interactions in the MAGICAL database to facilitate the research. MAGICAL-DB is a one-stop portal that allows users to identify genetic interactions for their genes of interest. The database offers different functionalities, such as a user can input the gene of interest, a pair of genes together, and also a set of genes or pairs for which the genetic interactions are to be identified. MAGICAL helps determine whether a gene pair is SL, SV, or NOT a genetic interaction. The graphical user interface lets users browse the

database and identify and explore genetic interactions. The database is available at http://sls.uohyd.ac.in/new/magicaldb.

## Discussions

In the present work, we have developed MAGICAL, a multi-class classifier model that identifies whether or not a given gene pair can genetically interact. There are more SL pairs than SVs identified and reported to date in the literature and databases. Identification of SL/SV pairs is challenging with statistical models, as they are not suitable for lower sample sizes and are based on hypothesis testing. Since in tumor samples, most of the genes are altered/mutated, it is hard to decipher the alteration of which two genes leads to an SV effect. If an alteration in two genes co-occurs, it does not necessarily mean the interaction involved is SV. It also could be that both genes are oncogenes or poor prognostic genes, so simultaneous alteration of both genes can show better progression of tumor cells and poor survival of patients. Another probable reason could be the target selectivity. In the case of SL interaction, targeting the partner of a mutant tumor suppressor gene leads to cell lethality; similarly, in SDL interaction, targeting the partner of a mutant oncogene leads to cell lethality. Such conclusions are yet to be drawn for the identification of SV interactions. The network topology and how these pairs are placed in the physical protein-protein interaction network would enable the understanding of SV interactions and which gene may rescue the effect and lead to resistance in the biological network.

The existing machine learning models to predict genetic interactions are all binary classifiers and mostly predict SL interactions. Although these binary classifiers can successfully predict SL, but fail in the following contexts. (i) These models are restricted to identifying SL and NOT classes where the NOT class consists of other genetic interactions such as SVs, SDLs, etc. (ii) Most of the models are not interpretable and are trained on limited datasets. In contrast, MAGICAL is trained on a more extensive and balanced dataset, can predict multiple types of genetic interactions, and is interpretable.

Previous independent studies report that SV interactions share more biological processes, and SL pairs share fewer biological processes [21,25]. In 2023, Liu *et al*., built a random forest classifier to predict SV interactions [21]. The model is trained on 220 SV and 220 non-SV gene pairs from CRISPR/Cas9 genetic screens. Features such as paralog gene, shared protein-protein interactors, similarity of biological process, protein complex membership, etc., are utilized. They observe that the SV pairs share more biological processes and have a higher essentiality of protein complex memberships (protein complexes including both geneA and geneB). Another study by Benstead-Hume *et al*. claims that SL pairs share fewer biological process GO terms and are located at the peripheries of communities connecting respective clusters [25]. These two independent findings corroborate our result that SL interactions share fewer biological processes and are located in different communities, whereas SV pairs share more biological processes and are part of the same protein complex or community.

MAGICAL and SLant are based on the topological properties of physical protein-protein interactions. Interestingly, both models identify pair-wise properties as better discriminators than node-wise properties. Among all the network properties, the shortest path is selected as one of the top-most discriminatory features for predicting genetic interactions. Adhesion is another pair-wise property selected by MAGICAL for classifying SL, SV, and NOT. Both adhesion and shortest path are significantly different for SV compared to SL pairs. The placement of SL pairs in different subnetworks/modules indicates mutual exclusivity. For example, if the BRCA1 gene in the first module undergoes inhibition and causes DNA damage, VEGFA (known to stimulate anti-apoptotic signals), the partner gene in the second module, repairs the

damage and keeps the cell viable. But if both BRCA1 and VEGFA genes are inhibited, the damage cannot be repaired, and the cell undergoes lethality.

Binary classifiers such as SLant and one built by Liu *et al.* report lower shortest paths for SL pairs and higher shortest distances for SV pairs, respectively, whereas MAGICAL reports that SL interactions are significantly higher than the SVs [21,25]. The disparity between these models and MAGICAL might be due to the following reasons. First, the training dataset for all the three models is different; both the binary classifiers have been trained on a minimal dataset, whereas MAGICAL has been trained on extensive datasets; second, the binary models are restricted to the identification of SL vs non-SL, and SV vs non-SV, MAGICAL, on the contrary, is trained on both positive and negative genetic interaction data, and enables the prediction of SL, SV, and NOT pairs; third, the NOT dataset for MAGICAL is picked randomly unlike the two binary classifiers.

Among the discriminatory properties, the betweenness of a node depicts the number of shortest paths crossing it, making it a bridge that establishes communication between two modules/communities. SL pairs have a higher value of "average betweenness" than the SV pairs, again representing that the SL pairs are more central in the network than the SVs. Triangles denote the extent to which the nodes in the network cluster together. SL interactions tend to cluster together and are densely connected compared to the SVs. We also found that SL pairs have a higher value of average neighbor2 than those of SV pairs, which conveys that SL pairs are connected or share a large number of neighbors and engage in more crosstalk. SL pairs have higher average betweenness, average triangle, and average neighbor2 values than SVs. In conclusion, SLs are present in different modules with high centrality measures, whereas SVs are present in the same/similar module with low centrality measures.

MAGICAL, though a robust machine learning model, still has scope for improvement. This study utilizes topological properties of the physical protein-protein interaction network. Although these properties are sufficient to understand the differences in genetic interactions, incorporating more features might contribute to better learning and comprehension. The current version of MAGICAL has three prediction classes: SL, SV, and NOT. More classes can be added to the model, such as synthetic dosage lethality, collateral lethality, etc.

In the future, models can be developed to predict higher-order genetic interactions, which are interactions between more than two pairs of genes. As the number of pairs would be enormous, achieving it would be laborious, time-consuming, and experimentally expensive. Few studies have identified trigenic interactions [41–43]. The expression/alteration of the third gene can affect the phenotype of SL or SV pairs. Thus, generating a computational pipeline would be beneficial to understand such pairs better. It has been reported that genetic interactions tend to show duality in their phenotype. Different research groups have identified that these genetic interactions undergo phenotype switching. Studies by Xianghua Li *et al.* and Xia Ding *et al.* show cases where the same pair can be both an SL and SV interaction depending on the context [44,45]. For example, Knockdown of PARP1 followed by BRCA1 inhibition leads to cell viability, and deletion of BRCA1 followed by PARP1 inhibition leads to cell lethality. The study by Magen *et al.* demonstrates the activation of positive interactions in some cancer tissues (breast and lung), while in other tissues, there is the activation of negative interactions [46]. The identification of context-specific genetic interaction has also not been explored. Identifying such pairs might be a notable discovery in the research arena of genetic interaction.

## Supporting information

**S1 Fig. Barplot showing frequency of model accuracy for 1000 random NOT datasets.**
(TIFF)

**S2 Fig. Barplot depicting accuracy metrics for average, difference, maximum, minimum, and summation of network properties.**
(TIFF)

**S3 Fig. Barplot representing accuracy metrics for different machine learning models.**
(TIFF)

**S4 Fig. Variable importance plot that illustrates the importance of each feature.** Note: The importance of the feature decreases from top to bottom.
(TIFF)

**S5 Fig. A stacked barplot representing the determinant features for 100 bootstraps.**
(TIFF)

**S6 Fig. The correlation plot depicting the correlation among the different network properties.** (Blue: positive correlation, Red: negative correlation). The size of the circles indicates the significance of the p-value for spearman correlation test.
(TIFF)

**S7 Fig. Barplot shows drop in accuracy when one of the features among average betweenness, average neighbor2, and average triangle is removed from feature list.**
(TIFF)

**S8 Fig. The AUC/ROC plot for more than one running point, different cutoffs/thresholds of 0.1–0.9 (from A to I) have been considered.** For each plot, we notice a lower value of False Positive Rate and a higher value of True Positive Rate.
(TIFF)

**S9 Fig. The error decreases to 100 trees and later becomes saturated.**
(TIFF)

**S10 Fig. Comparing AUC values of the multi-class model against three binary class models on unseen DepMap dataset (p-value 6.98e-06, 2.2e-16, and 2.2e-16, respectively, DeLong test).**
(TIFF)

**S11 Fig. Barplot representing the fraction of SL and SV pairs in different and same communities.** Most of the SL and SV pairs belong to different communities, and there is a difference in the proportion of SL and SV in different communities (p-value 2.2e-16, two-proportions z-test).
(TIFF)

**S1 Table. Novel SL pairs predicted by MAGICAL for the CRISPR dataset.**
(XLS)

**S2 Table. Novel SV pairs predicted by MAGICAL for the CRISPR dataset.**
(XLS)

## Acknowledgments

We want to thank Surabhi Kavya Sri, project IMSC Systems Biology, who worked on the preliminary results, and Haneesh Jindal, SERB-project staff, whose valuable suggestions were critical to the project's building.

## Author Contributions

**Conceptualization:** Manjari Kiran.

**Data curation:** Anubha Dey.

**Formal analysis:** Anubha Dey.

**Funding acquisition:** Manjari Kiran.

**Investigation:** Anubha Dey, Manjari Kiran.

**Methodology:** Anubha Dey, Manjari Kiran.

**Project administration:** Manjari Kiran.

**Resources:** Anubha Dey, Suresh Mudunuri.

**Software:** Anubha Dey, Manjari Kiran.

**Supervision:** Manjari Kiran.

**Validation:** Anubha Dey.

**Visualization:** Anubha Dey, Suresh Mudunuri, Manjari Kiran.

**Writing – original draft:** Anubha Dey.

**Writing – review & editing:** Anubha Dey, Manjari Kiran.

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
