## [Decision Letter · Decision Letter 0]

24 Apr 2024

Dear Dr. Kiran,

Thank you very much for submitting your manuscript "MAGICAL: A multi-class classifier to predict synthetic lethal and viable interactions using protein-protein interaction network" for consideration at PLOS Computational Biology.

As with all papers reviewed by the journal, your manuscript was reviewed by members of the editorial board and by several independent reviewers. In light of the reviews (below this email), we would like to invite the resubmission of a significantly-revised version that takes into account the reviewers' comments.

In particular, the reviewers raise concerns regarding bootstrapping and overfitting. Additionally, care must be taken to refine the flow of the introduction and improve the methods.  

We cannot make any decision about publication until we have seen the revised manuscript and your response to the reviewers' comments. Your revised manuscript is also likely to be sent to reviewers for further evaluation.

Sincerely,

Mohammad Sadegh Taghizadeh, Ph.D.

Academic Editor

PLOS Computational Biology

Stacey Finley

Section Editor

PLOS Computational Biology

Reviewer's Responses to Questions

**Comments to the Authors:**

Reviewer #1: 

The work is very good, simple, direct and with an important contribution to the area. I just want to note that when showing the ROC curves, it is important to indicate the confidence intervals of the AUC values and, if possible, perform a DeLong test to compare them and indicate significant differences,and then discuss the results, taking into consideration these differences demonstrated by formal statistical methods. Because the ROC method usually uses bootstraps, confidence intervals can be obtained by randomizing the bootstrap.

Reviewer #2: 

Review uploaded as an attachment.

Reviewer #3: 

The authorss propose a model named MAGICAL, a multi-class machine learning model, predicts SL and SV using network properties of proteins. It achieves ~80% accuracy on training data and performs well on experimental datasets. The work has potentials, however, I have the following suggestions/comments:

-AUCROC must be calculated/plotted by more than one running point(TPR,FPR). The one in the manuscript only have one point.

-some parameters values you need to mention why selected them e.g. why 100 bootstraps? justification is needed.

-How the model avoided overfitting. The authors may plot the training versus validation accuracy versus multiple running points (epochs) and check.

-The survival plots (D and E), the difference between the 2 curves does not seem to be significant. I wish that the CI at each time-point is shown or at least the p-value.

-KEGG pathway analysis and go-enrichment can be used to analyze the findings.

- The introduction uses suggest and suggested, I wish to unify the grammar.

**Have the authors made all data and (if applicable) computational code underlying the findings in their manuscript fully available?**

Reviewer #1: Yes

Reviewer #2: Yes

Reviewer #3: Yes

PLOS authors have the option to publish the peer review history of their article (what does this mean?). If published, this will include your full peer review and any attached files.

Reviewer #1: No

Reviewer #2: No

Reviewer #3: **Yes: **Abedalrhman Alkhateeb
---

## [Decision Letter · Decision Letter 1]

11 Jul 2024

Dear Dr. Kiran,

Thank you very much for submitting your manuscript "MAGICAL: A multi-class classifier to predict synthetic lethal and viable interactions using protein-protein interaction network" for consideration at PLOS Computational Biology.

As with all papers reviewed by the journal, your manuscript was reviewed by members of the editorial board and by several independent reviewers. In light of the reviews (below this email), we would like to invite the resubmission of a significantly-revised version that takes into account the reviewers' comments.

In particular all points raised by Reviewer 2, including issues regarding statistical analyses and refining of the figures, should be addressed.

We cannot make any decision about publication until we have seen the revised manuscript and your response to the reviewers' comments. Your revised manuscript is also likely to be sent to reviewers for further evaluation.

Sincerely,

Mohammad Sadegh Taghizadeh, Ph.D.

Academic Editor

PLOS Computational Biology

Stacey Finley

Section Editor

PLOS Computational Biology

Reviewer's Responses to Questions

**Comments to the Authors:**

Reviewer #2: While the paper has significantly improved from the initial review, there is still lacking information that is critical for understanding.

1. The train/test split is still not fully explained. I understand that SLant was trained on the 411 SL and 411 non-SL, but what was MAGICAL trained on? How many samples in the training data? How many samples in the test data?

2. Was SLant employed on the data used in this study? Or were the predictions taken from the cited papers?

3. Why were Kaplan Meier curves only plotted using TCGA data and not using the training dataset?

4. Statistical Analyses section needs to be further expanded explain why/how tests are chosen. As stands, it is lacking information to reproduce results.

Figures:

1. In general, make sure titles are on plots where it is not readily apparent what the graph is showing.

2. Figure 5: it would be helpful to display p-values in KM plot similarly to AUCs in ROC plots.

3. Figure 6: node colors should be consistent with the colors used in other plots

4. Supp. Figure 2: x-axis label is not correct. Please adjust.

Supplemental tables 1 and 2 do not appear in the supplemental document (if submitted as separate files, disregard).

Reviewer #3: The authors addressed the reviewers concerns. The manuscipr is in a very good shape for publication.

**Have the authors made all data and (if applicable) computational code underlying the findings in their manuscript fully available?**

Reviewer #2: Yes

Reviewer #3: None

PLOS authors have the option to publish the peer review history of their article (what does this mean?). If published, this will include your full peer review and any attached files.

Reviewer #2: No

Reviewer #3: **Yes: **Abedalrhman Alkhateeb
---

## [Decision Letter · Decision Letter 2]

17 Jul 2024

Dear Dr. Kiran,

We are pleased to inform you that your manuscript "MAGICAL: A multi-class classifier to predict synthetic lethal and viable interactions using protein-protein interaction network" has been provisionally accepted for publication in PLOS Computational Biology.

Before your manuscript can be formally accepted you will need to complete some formatting changes, which you will receive in a follow up email. A member of our team will be in touch with a set of requests.Please note that your manuscript will not be scheduled for publication until you have made the required changes, so a swift response is appreciated.

Should you, your institutions press office or the journal office choose to press release your paper, you will automatically be opted out of early publication. We ask that you notify us now if you or your institution is planning to press release the article. All press must be co-ordinated with PLOS.

Best regards,

Mohammad Sadegh Taghizadeh, Ph.D.

Academic Editor

PLOS Computational Biology

Stacey Finley

Section Editor

PLOS Computational Biology

Reviewer's Responses to Questions

Comments to the Authors:Please note here if the review is uploaded as an attachment.

Reviewer #2: The authors have made all requested edits. Their clarifications help for reader understanding.